# Non-Destructive Sensing of Tea Pigments in Black Tea Rolling Process

**DOI:** 10.3390/foods14213723

**Published:** 2025-10-30

**Authors:** Xuan Xuan, Ting An, Hanting Zou, Jiancheng Ma, Yongwen Jiang, Haibo Yuan, Haihua Zhang

**Affiliations:** 1College of Food and Health, Zhejiang A&F University, Hangzhou 311300, China; 18334363220@163.com; 2National Key Laboratory for Tea Plant Germplasm Innovation and Resource Utilization, Tea Research Institute, The Chinese Academy of Agricultural Sciences, Hangzhou 310008, China; anting_mac@163.com (T.A.);; 3School of Tea and Food Science and Technology, Anhui Agricultural University, Hefei 230036, China

**Keywords:** black tea rolling, electrical characterization, tea pigments, chemometrics, non-destructive sensing

## Abstract

Rolling is a critical step in the processing of black tea, marking the beginning of fermentation. At this stage, the formation of tea pigments causes significant changes in the color of the processed leaves, laying the essential groundwork for the development of color and flavor quality components in subsequent fermentation processes. However, the rapid and non-destructive sensing of tea pigments during black tea rolling remains challenging. This study focused on black tea products undergoing rolling as its research subject, utilizing electrical characteristic detection technology to collect time-series electrical parameters of rolling leaves at various testing frequencies. The original electrical parameters were preprocessed using multiplicative scatter correction (MSC), min-max normalization (Min-Max), and smoothing (Smooth). Various selection methods, including the competitive adaptive reweighting algorithm (CARS), uninformative variable elimination (UVE), and the variable combination population analysis and iterative retained information variable algorithm (VCPA-IRIV), were employed to identify electrical parameters relevant to the targeted attributes. Quantitative prediction models for the content of tea pigments were established using partial least squares regression (PLSR) and support vector machine regression (SVR). The results demonstrated that the Smooth-VCPA-IRIV-SVR model exhibited superior performance in predicting the contents of theaflavins (TFs), thearubigins (TRs), and theabrownins (TBs). Correlation coefficients of prediction (R_p_) all exceeded 0.99, and Relative prediction deviation (RPD) values were all above 6.5, indicating that the model enables rapid and non-destructive detection of tea pigment content during black tea rolling. These findings provide preliminary technical support and reference for the digital production of black tea.

## 1. Introduction

Tea is highly valued for its rich nutritional composition and significant health benefits [1]. Among the various types, black tea is renowned not only for its distinctive flavor and mellow taste but also for health-promoting functions, such as antioxidant, antihypertensive, and blood lipid-glucose regulatory activities. These attributes have solidified its status as the most widely traded tea category globally [2]. Rolling is a crucial primary processing step that disrupts tea leaf cells through mechanical force. This enables contact between polyphenol oxidase and its substrates, thereby accelerating the enzymatic oxidation of polyphenols. This process leads to the formation of water-soluble oxidation products, including theaflavins (TFs), thearubigins (TRs), and theabrownins (TBs) [3]. Additionally, rolling is considered the initial phase of fermentation [4,5]. At this stage, the color of the leaves changes from green to yellowish-green and then to light reddish-brown, mainly because of the production of pigment components: orange-yellow TFs, reddish-brown TRs, and brown TBs [6]. The formation and accumulation of these pigments establish the essential material basis for developing color- and flavor-related quality components in the subsequent fermentation stage [3,4,5,6,7]. Given the important role of tea pigments during rolling, they were selected as key physicochemical indicators for monitoring the black tea rolling process.

In the traditional processing of black tea, despite the alteration in the color of the tea leaves during rolling, tea masters cannot precisely quantify the content of tea pigments through visual inspection alone [8]. Accurate determination of TFs, TRs, and TBs during the rolling process requires sophisticated instruments and standardized detection methods for processing and measuring time-series rolling samples [9,10,11]. However, these methods typically involve complex sample preparation, lengthy procedures, and reliance on costly equipment and specialized operators, making it difficult to achieve online monitoring and real-time feedback on pigment content during rolling. Consequently, developing a rapid, efficient, and accurate method to assess tea pigment content is of significant importance for enhancing detection efficiency and advancing the digital processing of black tea.

In recent years, non-destructive testing technology has been widely used in agricultural products. However, research on non-destructive sensing of key physicochemical indicators during the black tea rolling stage remains relatively limited. Most existing studies have focused on non-destructive assessment of tea pigment content using machine vision technology [3,7]. Zou et al. analyzed the correlation between color features and tea pigment content in sequential rolling samples, confirmed the significant contribution of tea pigments to rolling quality, and established a quantitative prediction model for tea pigments based on visual image features [3]. Shen et al. employed machine vision combined with artificial intelligence algorithms to qualitatively determine the optimal rolling stage and developed a quantitative sensing model for TFs and their monomers, enabling accurate and non-destructive monitoring of tea pigment content during rolling [7]. Although machine vision shows considerable potential in assessing pigment-related components, it relies mainly on surface information and has limited penetration depth, making it difficult to directly quantify internal characteristics. In contrast, electrical characteristic-detection technology can overcome these limitations by providing a comprehensive digital representation of the entire rolled leaf. Notably, it functions by monitoring changes in intracellular electrical parameters (e.g., inductance, capacitance, resistance) to reflect the dynamic changes in internal chemical constituents during rolling [8,12]. As rolling proceeds, changes in cellular structure and internal composition affect charge distribution, thereby changing the electrical parameters of the leaves. Therefore, by analyzing the variations in electrical parameters across different test frequencies, the changes in tea pigments can be indirectly characterized. This technology has already demonstrated unique advantages during the fermentation stage of black tea processing. Zhu et al. established a qualitative discrimination model based on the correlation between electrical parameters and tea pigments, developing a rapid classification method for fermentation degree using the electrical parameters of the fermenting tea leaves [8]. Dong et al. combined electrical parameters with chemometric methods to construct a linear dynamic prediction model for catechin content during fermentation, enabling rapid and non-destructive detection of catechins [12]. An et al. proposed a fusion strategy integrating hyperspectral imaging and electrical characteristics, using stacked multi-source data to establish quantitative models for total catechins, soluble sugars, and caffeine during fermentation [13]. Due to its strong penetration ability and broad detection range, electrical characteristic-detection technology has been widely used in black tea fermentation. However, its application during the rolling process has not been reported. Therefore, introducing this technology into detection during the rolling process holds significant application potential.

In summary, this study targets tea pigments during black tea rolling and innovatively applies electrical characteristic-detection technology to collect seven electrical parameters from sequential rolling samples at thirty different test frequencies. Artificial intelligence algorithms were used to select feature frequencies and electrical parameters correlated with tea pigment content. Based on these features, both linear (PLSR) and nonlinear (SVR) quantitative prediction models were developed and compared to achieve rapid, non-destructive sensing of tea pigments during rolling. The specific steps of the study are as follows: (1) Preparation of sequential black tea rolling samples. (2) Collection of sample electrical characteristic parameter information. (3) Accurate determination of tea pigment content. (4) Data preprocessing and screening characteristic variables. (5) The prediction model of tea pigment based on electrical characteristic detection technology was established and compared. The overall research approach and technical route are illustrated in Figure 1.

## 2. Materials and Methods

### 2.1. Sample Preparation and Experimental Procedure

The tea cultivar used in this study was ‘Fuding Dabaicha’. Fresh leaves with a minimum of one bud and two leaves were plucked on the afternoon of 12 April 2024 from the Shengzhou Experimental Base of the Tea Research Institute, Chinese Academy of Agricultural Sciences. The leaves were processed following the black tea Processing Technical Specification (GB/T 35810-2018 [14]). Fresh tea leaves were spread in a withering tank at a thickness of 2 cm. Under 24 °C temperature and 60% relative humidity, withering took 12 h until moisture content reduced from 77.6% to 61.1%. The withered leaves were then subjected to rolling using a rolling machine (6CR-45, Xiangfeng Machinery Co., Ltd., Tongxiang, China). The rolling process lasts 90 min in total, involved alternating sequences of no pressure, light pressure, and heavy pressure, as follows: no pressure (12 cm gap) for 20 min → light pressure (18 cm gap) for 20 min → heavy pressure (25 cm gap) for 10 min → light pressure (18 cm gap) for 10 min → heavy pressure (25 cm gap) for 10 min → light pressure (18 cm gap) for 20 min. The process was dominated by light rolling, with shorter periods of heavy pressure. Samples were collected every 10 min, resulting in 10 sampling time points. In all, 15 parallel samples were taken at each time point, yielding in a total of 150 rolling samples. After measuring the electrical characteristics, each sample was immediately frozen in liquid nitrogen and stored for subsequent freeze-drying and analysis of tea pigment content.

### 2.2. Acquisition of Electrical Parameters

The electrical parameter detection system for rolled tea leaves comprised an LCR meter (TH2829C, Tonghui Electronic Co., Ltd., Changzhou, China) and a custom-built parallel-plate electrode chamber. The electrodes were made of red copper, and the chamber was constructed from insulating PMMA. Before measurement, the LCR meter was warmed up for 30 min. Each of the 150 rolled tea samples was placed in the sample chamber, and their electrical parameters were measured at different test frequencies using the control software. Each sample was measured three times, and the average value of each parameter was calculated and recorded as the final result. To ensure sufficient electrode–sample contact, a consistent pressure was applied using an insulating glass plate placed on top of the sample during each measurement. The LCR meter was set to a test voltage of 1 V. Measurements were conducted at 30 characteristic frequencies ranging from 0.02 kHz to 100 kHz: 0.02, 0.04, 0.06, 0.08, 0.1, 0.15, 0.2, 0.3, 0.4, 0.5, 0.6, 0.7, 0.8, 0.9, 1, 2, 4, 6, 8, 10, 15, 20, 30, 40, 50, 60, 70, 80, 90, and 100 kHz. The seven measured electrical characteristic parameters were: series equivalent inductance (Ls), series equivalent capacitance (Cs), series equivalent resistance (Rs), complex impedance (Z), reactance (X), dissipation factor (D), and quality factor (Q). Consequently, a two-dimensional data matrix of 150 × 210 (30 test frequencies × 7 parameters) was obtained for the entire rolling process.

### 2.3. Determination of Tea Pigment Content

The contents of tea pigments (TFs, TRs, and TBs) were quantitatively determined using a systematic analytical method based on extraction with organic solvents (ethyl acetate, n-butanol, and ethanol) followed by ultraviolet spectrophotometry, according to established methodologies [3,9,10,11]. All biochemical analyses were performed in triplicate to ensure reliability and accuracy, and the final reported values represent the mean of the three replicates.

Preparation reagent: 2.5% sodium bicarbonate solution, saturated oxalic acid solution.

Preparation of mother liquor: A 3 g tea sample was weighed into a 250 mL conical flask, mixed with 125 mL of boiling water, extracted in a 100 °C boiling water bath for 10 min, and then filtered and cooled.

Separation and purification: A 30 mL aliquot of tea infusion was mixed with an equal volume of ethyl acetate and shaken for 5 min. The aqueous layer was used to prepare solution D (2 mL saturated oxalic acid solution, 6 mL pure water, made up to 25 mL with 95% ethanol). Then, 2 mL of the ethyl acetate layer was used to prepare solution A (made up to 25 mL with 95% ethanol). Another 15 mL of the ethyl acetate layer was taken, mixed with an equal volume of sodium bicarbonate solution, and shaken for 30 s. Subsequently, 4 mL of the ethyl acetate layer was taken to prepare solution C (made up to 25 mL with 95% ethanol). Separately, 15 mL of tea infusion was mixed with an equal volume of n-butanol, shaken for 3 min, and allowed to separate. The aqueous layer was used to prepare solution B (2 mL saturated oxalic acid solution, 6 mL pure water, made up to 25 mL with 95% ethanol).

Colorimetric determination: Using an ultraviolet spectrophotometer and a 1 cm quartz cuvette, with 95% ethanol as the blank for zeroing, the absorbance values of solutions A, B, C, and D were measured at 380 nm and recorded as *EA*, *EB*, *EC*, and *ED*. The content of each component was calculated using the following formulas:
(1)TF%=EC×2.25
(2)TR%=(2EA+2ED−2EB−EC)×7.06
(3)TB%=2EB×7.06

### 2.4. Data Processing and Analysis

#### 2.4.1. Data Preprocessing

The electrical parameters were standardized to transform data with different measurement units into a uniform scale [15]. Three methods were employed: MSC, Min-Max, and Smooth [16].

#### 2.4.2. Feature Variable Selection

Correlations exist among the full-spectrum frequency variables, meaning a variable at one frequency point can be jointly explained by variables at other points. Such inter-correlation introduces data redundancy, which may compromise model predictive ability. To extract feature variables relevant to tea pigment content from the multi-frequency electrical parameters, three variable selection algorithms were applied: CARS [17], UVE [18], and VCPA-IRIV [11].

The principles of these algorithms are as follows: (1) CARS: This algorithm employs the absolute values of PLS regression coefficients as its evaluation metric. The process initiates with Monte Carlo sampling, which randomly selects a subset of the training set to build a PLSR model. It then iteratively refines the variable subset by applying an exponential decay function and adaptive reweighted sampling based on the regression coefficients. Finally, the subset of features yielding the smallest root mean square error of cross-validation (RMSECV) is identified and selected as the final output of the algorithm [17,19]. (2) UVE: This algorithm adds random noise variables to the electrical parameter matrix. A PLSR model is built using leave-one-out cross-validation, generating a regression coefficient matrix. The stability of each variable is calculated as the ratio of the mean of its regression coefficients to their standard deviation. Variables whose stability exceeds a certain threshold are retained as informative features [18]. (3) VCPA-IRIV: This is a hybrid variable selection strategy combining VCPA and IRIV to leverage their respective advantages. VCPA rapidly reduces the variable space using an exponential decay function and binary matrix sampling over multiple iterations, yielding a preliminary variable subset. IRIV then further assesses the importance of these remaining variables, considering interactions among them [20]. It eliminates uninformative and interfering variables, retaining only those most relevant to tea pigment content.

#### 2.4.3. Model Development and Evaluation

Modeling was performed using MATLAB (Version 2022a, MathWorks, Natick, MA, USA) with two distinct algorithms. PLSR, a multivariate statistical regression technique, integrates principal component analysis and multiple regression. It excels at managing multicollinearity in data by extracting latent variables that maximize the covariance between predictor and response variables, thereby optimizing model performance. SVR, based on support vector machine theory, captures nonlinear relationships. It operates by mapping input data into a higher-dimensional feature space through a nonlinear transformation and performing regression by minimizing the margin around the support vectors. The SVR model’s flexibility allows for the selection of different kernel functions and hyperparameters to enhance predictive accuracy. In this study, a nonlinear SVR model was constructed using the Radial Basis Function (RBF) kernel. The kernel parameter (g) and the penalty factor (c) were optimized through a grid search combined with cross-validation [21].

Model performance was evaluated using four key metrics: the correlation coefficient of the cross-validation set (R_cv_), R_p_, and cross-validation set (RMSECV), the prediction set (RMSEP) [22,23,24]. Generally, models with R_cv_ and R_p_ values closer to 1 and lower RMSECV and RMSEP values indicate superior predictive capability. Additionally, RPD was used as a critical evaluation criterion. An RPD value greater than 2 suggests that the model possesses satisfactory predictive ability [13]. A 10-fold cross-validation was applied to all samples to ensure model robustness and reliability.

## 3. Results and Discussion

### 3.1. Variation Trends in Electrical Parameters

Figure 2a–g illustrate the variation trends in the electrical parameters by testing frequency at different rolling times. As shown in Figure 2a,b, Ls and Cs exhibited opposite trends: Ls increased with frequency, while Cs decreased, with both showing the most pronounced variations in the low-frequency range. Additionally, both Ls and Cs displayed an overall increasing trend as rolling time progressed. According to Figure 2c,f, although the absolute values of Rs and D differed, their variation patterns were generally consistent, both decreasing gradually with increasing frequency. As rolling time extended, Rs and D values showed a decreasing trend, with more evident changes in the low-frequency region. From Figure 2d,e,g, Q and Z initially increased and then decreased with frequency, whereas X exhibited an opposite trend, decreasing first and then increasing. With prolonged rolling time, Q and Z increased, while X decreased. Variations in Q and Z were more substantial at low-to-medium frequencies, while changes in X were more significant at higher frequencies. However, a notable inflection point in the trends of these electrical parameters occurred after 70 min of rolling. This phenomenon is likely associated with biochemical and physical changes resulting from tea cellular disruption under continuous mechanical stress. Prolonged rolling may alter polyphenol oxidase activity, reduce catechin and flavonoid content, and cause severe cellular damage leading to excessive tea juice exudation, all of which can collectively influence the electrical characteristics of the leaves [4]. In summary, testing frequency had a more pronounced effect on the electrical parameters of black tea during rolling than rolling time itself.

### 3.2. Variation Trends of Tea Pigments

Figure 3 shows the variation in TF, TR, and TB content in black tea samples at ten rolling time points, with rolling time on the *x*-axis and the percentage content of tea pigments on the *y*-axis. The visible color of the leaves gradually shifted from green to light reddish-brown as rolling progressed. the experimental data confirmed that the contents of TFs, TRs, and TBs increased with rolling duration. Specifically, TF content increased gradually, while TR and TB contents increased almost synchronously. This phenomenon occurs because the rolling process disrupts the leaf cell structure, allowing polyphenol oxidase to fully interact with polyphenolic substrates such as catechins. This initiates oxidation, forming TFs, which are subsequently oxidized and polymerized enzymatically to form TRs and TBs. These pigments significantly influence the ultimate color of black tea. Notably, the rates of change in the content of these three pigments differed throughout the rolling process, a variation potentially linked to the degree of cellular disruption and the activity of polyphenol oxidase [4,5].

### 3.3. Development of Predictive Models for Tea Pigment Content

#### 3.3.1. Optimization of Data Preprocessing Methods

The sample set of 150 was partitioned using the Kennard–Stone (K-S) algorithm based on Mahalanobis distance. A representative subset of 50 samples was selected as the prediction set, with the remaining 100 samples constituting the calibration set (a ratio of 1:2). A full-variable PLSR model was initially established using the original electrical parameters. As shown in Table 1, the RPD values for the prediction set of TF, TR, and TB contents were 3.329, 2.701, and 3.284, respectively, indicating reasonably good predictive performance. This suggests that the original parameters contain valid information relevant to the tea pigments. However, they also include interference, such as electronic noise from data acquisition and uneven particle size effects [24]. Therefore, the original data were preprocessed using MSC, Min-Max, and Smooth methods before constructing full-variable PLSR models for the tea pigments. The results in Table 1 show that only the Smooth-preprocessed data led to a higher RPD value for the prediction set. The RPD values of the prediction set in the prediction results of TFs, TRs, and TBs are 3.428, 2.856 and 3.609, respectively, which is the best preprocessing method. In contrast, both MSC and Min-Max preprocessing resulted in PLSR models with inferior performance compared to the model using raw data. This decline may be attributed to data distortion or loss of critical information during preprocessing, which reduces the signal-to-noise ratio and thus weakens predictive ability [25].

#### 3.3.2. Selection of Characteristic Electrical Parameters and Development of Linear Predictive Models

The seven electrical parameters (Ls, Cs, Rs, D, Q, Z, X), measured at 30 testing frequencies, were combined into a variable set of 210 dimensions. To improve computational speed and prediction accuracy, feature variables and frequencies correlated with tea pigment content were extracted from the Smooth-preprocessed data using the CARS, UVE, and VCPA-IRIV algorithms, aiming to build more concise and accurate prediction models. Before executing the CARS algorithm, key parameters were set as follows: Monte Carlo sampling runs = 60, and 5-fold cross-validation. For the UVE algorithm, the number of principal components was set to 10, with the threshold parameter at 0.995. For the VCPA-IRIV algorithm, the key parameters were configured as follows: the number of variables remaining after EDF = 100, BMS runs = 1000, iterations = 50, and variable importance was assessed using 5-fold cross-validation.

Figure 4 displays the feature variables selected by each algorithm, All three methods effectively compressed the variable space, achieving a compression rate exceeding 72%. For TFs, CARS, UVE, and VCPA-IRIV selected 20, 20, and 14 variables, respectively. Ls and X were identified as important features for characterizing TF variation, with low-frequency Cs and Rs also playing significant roles. For TRs, the algorithms selected 18, 57, and 10 variables, respectively, with Ls, Cs, and X being key features. For TBs, they selected 23, 25, and 13 variables, respectively. Notably, none of the algorithms selected identical parameters for a given pigment, indicating that each electrical parameter plays a unique role in characterizing the changes in tea pigments.

After Smooth preprocessing, feature variable PLSR models for predicting pigment content during rolling were established by integrating the three variable selection algorithms (CARS, UVE, VCPA-IRIV) with PLSR. The modeling results are summarized in Table 2. Comparison of PLSR models developed with different selection methods showed the following: (1) For TFs: The VCPA-IRIV algorithm yielded the optimal model. Compared to the full-variable PLSR model, the VCPA-IRIV-PLSR model increased R_p_ from 0.963 to 0.969 and decreased RMSECV from 0.052 to 0.047 and RMSEP from 0.056 to 0.052. The number of variables used for modeling was reduced from 210 to 14 (a 93% compression rate). The RPD of 3.559 indicated good predictive performance. Models based on CARS and UVE underperformed compared to the full-variable model. As suggested by Figure 4, the CARS algorithm might have eliminated important variables related to Q and Z at high frequencies, while the UVE algorithm’s selection exhibited discontinuous intervals, potentially leading to unrepresentative feature selection and reduced predictive ability. (2) For TRs: The VCPA-IRIV algorithm again performed best. The VCPA-IRIV-PLSR model increased R_p_ from 0.947 to 0.980, decreased RMSECV from 0.333 to 0.172, and RMSEP from 0.372 to 0.234. The variable count was reduced to 10 (95% compression). The high RPD of 4.660 indicated excellent predictive performance. All variable selection methods outperformed the full-variable model, successfully removing interfering noise while effectively capturing characteristic information. (3) For TBs: VCPA-IRIV remained superior. It increased R_p_ from 0.965 to 0.983, decreased RMSECV from 0.313 to 0.180, and RMSEP from 0.283 to 0.197, using only 25 variables (88% compression). The RPD of 5.347 signified very good predictive ability. Both the CARS and UVE models underperformed on the full-variable model, potentially because CARS removed important low-frequency Rs variables, and UVE selected redundant variables. Although the improvement in R_p_ for the best models (VCPA-IRIV-PLSR) might seem modest compared to the full-variable model, the significant reduction in the number of variables required for modeling drastically shortens computation time, which is crucial for the real-time sensing of tea pigments during rolling.

In summary, the VCPA-IRIV algorithm demonstrated the best overall performance for predicting tea pigment content. It effectively retained information highly relevant to the pigments, significantly enhancing the quantitative models. While CARS and UVE eliminated much redundant information, they also likely removed some important features related to pigment content, leading to decreased predictive performance. The fact that VCPA-IRIV generally selected fewer variables than CARS and UVE suggests the presence of significant multicollinearity within the original electrical parameters. A smaller set of feature variables is advantageous for reducing the cost and complexity of developing online monitoring equipment [26]. Notably, PLSR as a classical linear modeling technique, not only handles multicollinearity but also helps mitigate the influence of noise on the regression.

#### 3.3.3. Development of Nonlinear Predictive Models for Tea Pigment Content

However, the black tea rolling process involves complex biochemical reactions. The PLSR model can only capture linear relationships between variables and outcomes, potentially overlooking underlying nonlinear interactions [27,28]. To develop a more effective modeling method for the rapid, non-destructive detection of tea pigments, this study constructed a nonlinear SVR model based on the feature variables selected by the optimal VCPA-IRIV algorithm. As shown in Table 2, the SVR model demonstrated superior performance: for TFs, the prediction set yielded R_p_ = 0.995, RMSEP = 0.022, and RPD = 9.192; for TRs, R_p_ = 0.990, RMSEP = 0.168, RPD = 6.511; and for TBs, R_p_ = 0.995, RMSEP = 0.121, RPD = 9.135. The enhanced performance of the SVR model over PLSR suggests that the linear assumption in PLSR may be inadequate for fully characterizing the complex relationship between electrical parameters and tea pigments, whereas the nonlinear regression capability of SVR effectively captures these intricate correlations.

### 3.4. Model Comparison and Evaluation

Prediction accuracy is a key metric for evaluating model performance, but computational time is equally important for the real-time monitoring of black tea quality. Generally, the computational time required for prediction increases with the number of variables [13]. Therefore, a comprehensive model evaluation is necessary. Based on Smooth preprocessing, Figure 5 presents a bubble chart comparing the performance of three typical modeling approaches: the full-variable PLSR model, the VCPA-IRIV-PLSR model, and the VCPA-IRIV-SVR model. In the figure, red, green, and blue color schemes represent TFs, TRs, and TBs, respectively. Each scatter point corresponds to a model, positioned based on its R_p_ and RPD values from the prediction set. The point’s radius represents the model’s RMSEP value, and the annotated number indicates the count of variables used in the model. The superior performance model should have the following characteristics: away from the origin of the coordinate, the radius is small, and the number of variables is small. The results clearly show that the VCPA-IRIV-SVR model achieved the best performance for all three tea pigments. The Smooth preprocessing method enhances model robustness by reducing noise and random fluctuations. The VCPA-IRIV hybrid variable selection strategy effectively mitigates multicollinearity within electrical parameters through iterative variable space reduction, significantly improving model performance. This strategy yielded the most compact variable sets, eliminating a greater proportion of information irrelevant to the tea pigments while achieving the highest accuracy, making it highly suitable for real-time detection in practical production. Comparing linear and nonlinear models, the complex chemical changes during rolling imply that the relationship between electrical parameters and target pigments is not purely linear. The SVR algorithm, with its inherent nonlinear mapping capability and parameter optimization, demonstrated unique advantages in capturing the dynamic changes in pigment content [29].

Rolling is a critical step in black tea processing, profoundly impacting subsequent fermentation and final product quality. Previous studies by Zou et al. and Shen et al. mainly applied machine vision technology for quantitative assessment of tea pigments during rolling, reporting optimal model RPD values greater than 3.5 and 7.9 [3,7]. Although machine vision rapidly captures surface color and morphological features based on appearance and texture, it is susceptible to external environmental interference and has limited penetration depth, unable to detect internal sample characteristics [30]. Compared to machine vision, spectroscopic techniques using diffuse reflection probe slightly deeper but remain unsuitable for acquiring optical spectra inside an enclosed rolling drum [31]. Moreover, spectroscopy is affected by dynamic environmental conditions, and the collected spectra only represent localized information, compromising stability and comprehensiveness. In contrast, the electrical characteristic detection technology adopted in this study overcomes these limitations by acquiring more comprehensive digital information from rolled leaves. This study innovatively applied this technology for the non-destructive sensing of tea pigments during rolling, establishing quantitative prediction models for TFs, TRs, and TBs with remarkable RPD values of 9.192, 6.511, and 9.135, respectively. These results significantly outperform models based on machine vision. The superior performance may be attributed to the direct relationship between pigment formation and electrical properties. Mechanical rolling disrupts tea cells, releasing polyphenols that oxidize enzymatically to form pigments. The generation and dynamic changes in these pigments alter the electrical characteristics of the leaves. Electrical parameter acquisition provides a more holistic digital representation of samples compared to the surface-limited information from machine vision, resulting in enhanced predictive models. Additionally, sensing electrodes can be integrated into the rolling apparatus without structural modification, offering better application and promotion prospects than machine vision or spectroscopic techniques.

This study focused on the black tea rolling stage and constructed a quantitative prediction model for tea pigment content based on electrical characteristic detection technology. The feasibility of evaluating tea pigment content using electrical characteristics has been preliminarily explored, and the rapid and non-destructive perception of tea pigment in rolled products has been realized. However, certain limitations and future research directions should be noted. The current findings are based on a single geographical origin, harvest season, tea cultivar, and leaf maturity level. Their generalizability requires validation with more diverse samples. Given the complexity of tea and the dynamic changes in color, aroma, and composition during rolling, this study employed only electrical characteristic detection technology without incorporating other multi-source analytical methods. Future research will improve the generalizability and practicability of the model through transfer learning, that is, collecting representative samples in the target factory environment, fine-tuning the existing model, and quickly adapting the prediction model to the new production conditions with a small number of new samples, so as to lay a foundation for the online and rapid detection of tea pigments. Further integration of electrical characteristics with other multi-source analytical technologies will help construct a multi-dimensional tea quality evaluation system and promote the advancement of the tea industry toward high quality and digitalization.

## 4. Conclusions

To overcome the limitations of traditional methods that rely on subjective visual assessment and lack rapid biochemical detection, this study integrated electrical characteristic detection technology with chemometrics to develop a rapid and reliable analytical approach for estimating tea pigment content during rolling. Using three preprocessing methods, three variable selection algorithms, and two modeling techniques, a method for the quantitative and rapid detection of tea pigments was established. The results demonstrated the outstanding performance of the Smooth-VCPA-IRIV-SVR model, with all prediction set RPD values exceeding 6.5. This method facilitates the rapid and accurate quantitative analysis of tea pigment content during the rolling stage, providing a theoretical foundation for the precise processing and development of intelligent equipment for black tea. This study confirms the feasibility of quantitative evaluation of tea pigment content in black tea rolling process based on electrical characteristic detection technology and provided a theoretical basis for rapid and non-destructive detection of rolling quality, precise processing of black tea, and development of intelligent equipment.

## Figures and Tables

**Figure 1 foods-14-03723-f001:**
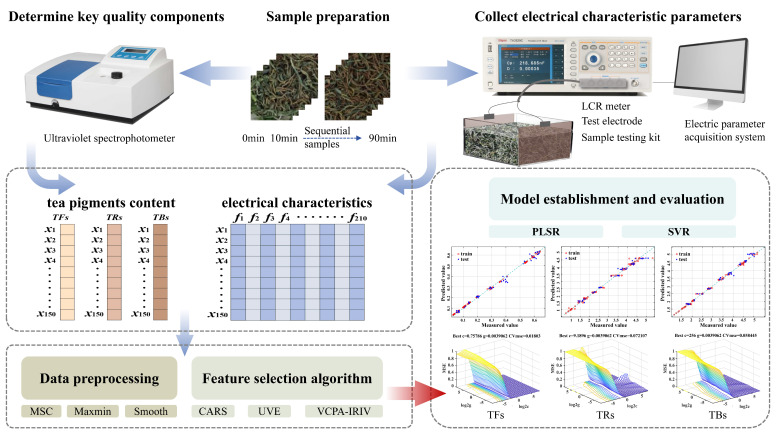
Technology Roadmap.

**Figure 2 foods-14-03723-f002:**
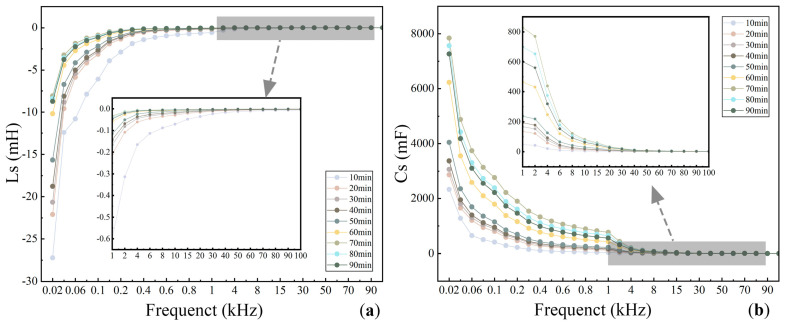
Electrical characteristic parameters of black tea as a function of test frequency. (**a**–**g**) are different electrical characteristic parameters. (**a**) represents Ls, series equivalent inductance, (**b**) represents Cs, series equivalent capacitance, (**c**) represents Rs, series equivalent resistance, (**d**) represents Z, complex impedance, (**e**) represents X, reactance, (**f**) represents D, dissipation factor, (**g**) represents Q, quality factor.

**Figure 3 foods-14-03723-f003:**
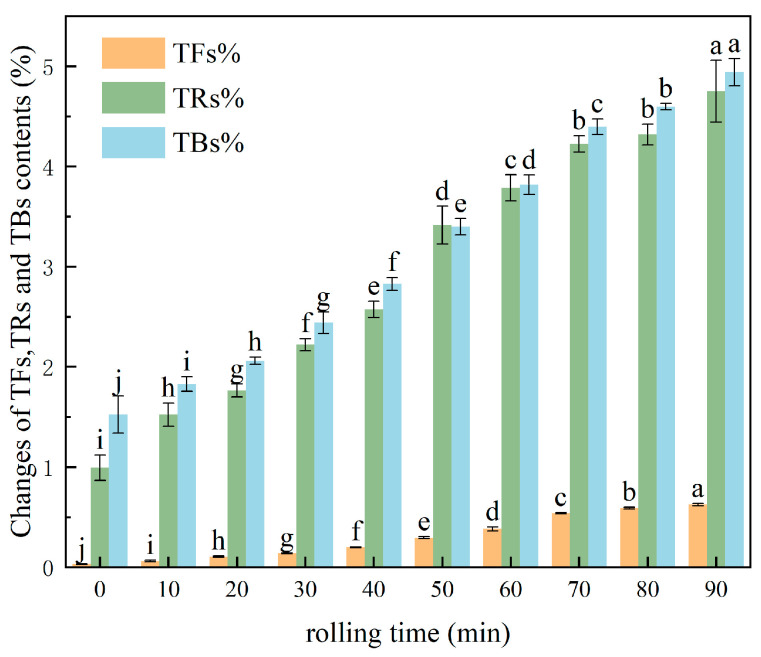
The change in tea pigment under different rolling times, yellow column represents TFs, green column represents TRs, blue column represents TBs. Values with different letters (a–j) in the same row indicated significant differences (*p* < 0.05).

**Figure 4 foods-14-03723-f004:**
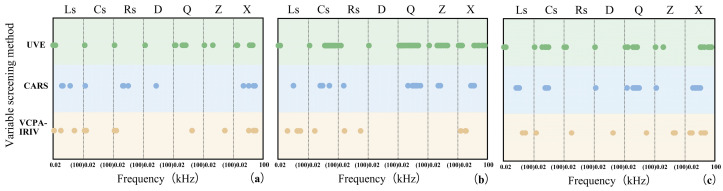
The characteristic electrical characteristic parameter variables selected by UVE (marked with green ⚪), CARS (marked with blue ⚪) and VCPA-IRIV (marked with yellow ⚪) were used to analyze (**a**) TFs, (**b**) TRs and (**c**) TBs.

**Figure 5 foods-14-03723-f005:**
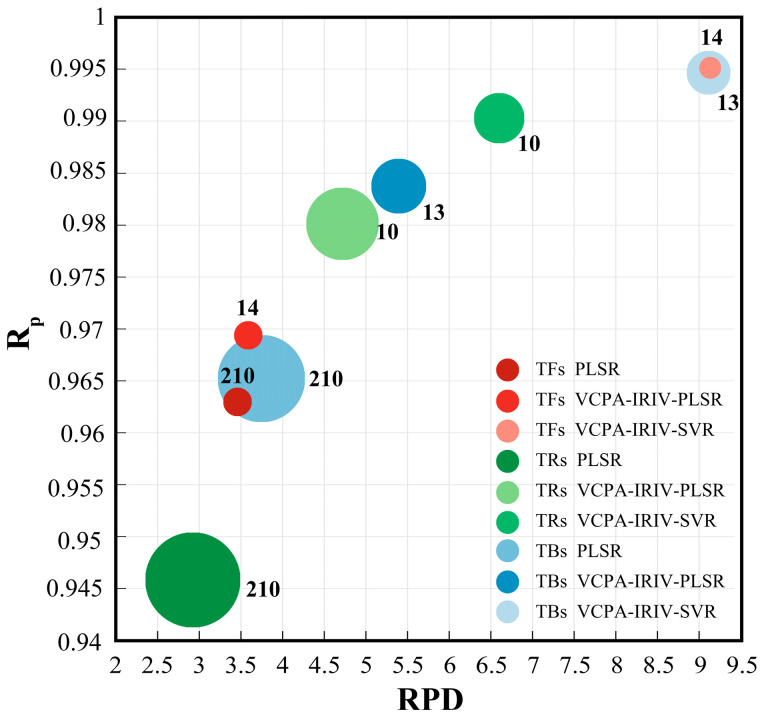
Comparison of the predictive performance of TFs, TRs and TBs based on the full-variable PLSR model, the VCPA-IRIV-PLSR model, and the VCPA-IRIV-SVR model. (Note: red represents TFs, green represents TRs, blue represents TBs, the number on ⚪ represents the number of selected variables, and the radius of ⚪ represents the RMSEP value).

**Table 1 foods-14-03723-t001:** Effect of different pretreatment methods on tea pigments in PLS model.

Quality Index	Preparation Method	PCs	Calibration Sets		Prediction Sets		
			R_cv_	RMSECV	R_p_	RMSEP	RPD
TFs	Raw	10	0.973	0.050	0.963	0.056	3.329
	MSC	10	0.977	0.042	0.929	0.070	2.429
	Min-Max	10	0.979	0.040	0.935	0.070	2.410
	Smooth	10	0.971	0.052	0.963	0.056	3.428
TRs	Raw	9	0.963	0.347	0.943	0.387	2.701
	MSC	10	0.986	0.201	0.917	0.398	2.268
	Min-Max	10	0.985	0.208	0.935	0.373	2.444
	Smooth	10	0.966	0.333	0.947	0.372	2.856
TBs	Raw	9	0.961	0.330	0.961	0.303	3.284
	MSC	10	0.978	0.227	0.936	0.342	2.555
	Min-Max	10	0.980	0.222	0.941	0.332	2.674
	Smooth	10	0.965	0.313	0.965	0.283	3.609

**Table 2 foods-14-03723-t002:** Comparison of tea rolling quality indexes predicted based on different feature extraction.

Model	Quality Index	Feature Extraction	Variables	PCs	Parameter	Calibration Sets		Prediction Sets		
						R_cv_	RMSECV	R_p_	RMSEP	RPD
PLSR	TFs	—	210	10	—	0.971	0.052	0.963	0.056	3.428
		CARS	20	7	—	0.976	0.044	0.964	0.055	3.362
		UVE	20	9	—	0.928	0.065	0.944	0.075	2.159
		VCPA-IRIV	14	8	—	0.976	0.047	0.969	0.052	3.559
	TRs	—	210	10	—	0.966	0.333	0.947	0.372	2.856
		CARS	18	9	—	0.983	0.211	0.965	0.307	3.471
		UVE	57	10	—	0.973	0.298	0.951	0.361	2.935
		VCPA-IRIV	10	9	—	0.991	0.172	0.980	0.234	4.660
		—	210	9	—	0.965	0.313	0.965	0.283	3.609
	TBs	CARS	23	9	—	0.976	0.260	0.961	0.296	3.351
		UVE	25	10	—	0.958	0.344	0.961	0.333	2.673
		VCPA-IRIV	13	10	—	0.989	0.180	0.983	0.197	5.347
SVR	TFs	VCPA-IRIV	14	—	c = 0.758 g = 0.004	1.000	0.010	0.995	0.022	9.192
	TRs	VCPA-IRIV	10	—	c = 9.190 g = 0.004	0.992	0.126	0.990	0.168	6.511
	TBs	VCPA-IRIV	13	—	c = 256 g = 0.004	0.999	0.059	0.995	0.121	9.135

## Data Availability

The original contributions presented in this study are included in the article. Further inquiries can be directed to the corresponding authors.

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
