# Peer review of "Non-Destructive Sensing of Tea Pigments in Black Tea Rolling Process"

_foods, 2025, doi:10.3390/foods14213723_

Round 1

Reviewer 1 Report

Comments and Suggestions for Authors

Overall, the study is interesting, timely and potentially useful for digital/automated black tea processing. The experimental concept and variety of chemometric approaches are strengths. However, methodological details, controls, mechanistic interpretation and statistical validation require strengthening before this work is suitable for publication in its present form.

The properties and health benefits of tea in general are required. Refer to https://doi.org/10.1016/j.chemosphere.2024.143550

L134: The authors state “a consistent pressure was applied using an insulating glass plate” — please quantify this (force or pressure), document how it was controlled and show that variations in applied pressure do not dominate the electrical signal. If a mechanical press with fixed weight was used, give mass; otherwise report standard deviation of contact pressure across replicates.

Dielectric and resistance measurements depend on sample temperature. State sample temperature at measurement (room temp? sample cooled/warmed after freezing?), whether measurements were done at controlled temperature, and whether temperature varied during the 90-min rolling campaign. If temperature varied, include temperature as covariate or show that it does not correlate with electrical parameters.

Provide full method parameters (extraction solvents, volumes, times, centrifugation speed, wavelengths used for TF/TR/TB quantification, calibration curves, standards). For TFs, TRs and TBs, UV methods can be semi-quantitative and prone to interference. Cite the exact method and show example calibration curves, linear ranges, LOD/LOQ and recovery (spike-and-recovery) results.

Discuss sensitivity of model to variation in initial raw material (leaf maturity, pre-withering moisture) and to processing parameters (rolling pressure regime). Could models be transfer-learned or require re-training for different factories?

Figure 3 is useful; legends should indicate units and include letters of statistical significance.

For all performance metrics report uncertainties (e.g., Rp ± SE) and p-values for key comparisons between models

Author Response

Comments 1: The properties and health benefits of tea in general are required. Refer to https://doi.org/10.1016/j.chemosphere.2024.143550 

Response 1: Thank you for your valuable suggestions, we have increased the characteristics and health benefits of general tea, in addition to focusing on the introduction of the research object-black tea.

Modify as follows (line38-42): Tea is highly valued for its rich nutritional composition and significant health benefits. Among them [1], Black tea is renowned not only for its distinctive flavor and mellow taste but also for a range of health-promoting functions, including antioxidant, antihypertensive, and lipid-glucose regulatory activities. These attributes have solidified its status as the most widely traded tea category globally [2].

Comments 2: L134: The authors state “a consistent pressure was applied using an insulating glass plate” — please quantify this (force or pressure), document how it was controlled and show that variations in applied pressure do not dominate the electrical signal. If a mechanical press with fixed weight was used, give mass; otherwise report standard deviation of contact pressure across replicates.

Response 2: Thank you for your valuable suggestions, in response to your problem, we did use a fixed quality insulating glass plate to apply pressure in the experiment. The glass plate has a size of 20 cm×10 cm and a mass of 5g. During the experiment, all 150 samples were placed by the same operator. The same glass plate was gently placed on the rolling leaves in the sample pool, and we ensured that the pressure conditions of all 150 samples were highly consistent.

Comments 3: Dielectric and resistance measurements depend on sample temperature. State sample temperature at measurement (room temp? sample cooled/warmed after freezing?), whether measurements were done at controlled temperature, and whether temperature varied during the 90-min rolling campaign. If temperature varied, include temperature as covariate or show that it does not correlate with electrical parameters.

Response 3: Thank you for your valuable suggestions, all measurements in this study were completed in a constant temperature laboratory, and the samples were always in a stable room temperature (about 25℃). During the entire 90-minute rolling process, the sample did not undergo any active temperature control, and its temperature was monitored to confirm that there was no significant change, which was consistent with the ambient temperature. Therefore, we believe that temperature is a controlled constant in the experiment, and its fluctuation has a negligible effect on the electrical signal, so it is not included in the analysis as a covariate.

Comments 4: Provide full method parameters (extraction solvents, volumes, times, centrifugation speed, wavelengths used for TF/TR/TB quantification, calibration curves, standards). For TFs, TRs and TBs, UV methods can be semi-quantitative and prone to interference. Cite the exact method and show example calibration curves, linear ranges, LOD/LOQ and recovery (spike-and-recovery) results.

Response 4: Thank you for your valuable suggestions, we add the complete method parameters.

Modify as follows (line168-187): The content of tea pigments (TFs, TRs, and TBs) was quantitatively determined using a systematic analysis method that involved extraction with organic solvents (ethyl acetate, n-butanol, and ethanol) followed by measurement with an ultraviolet spectrophotometer, according to established methodologies [7-9]. All biochemical analyses were performed in triplicate to ensure data reliability and accuracy, and the final reported values represent the mean of the three replicates.

Preparation reagent: 2.5 % sodium bicarbonate solution, saturated oxalic acid solution.

Preparation of mother liquor: 3g tea samples were weighed and added with 125mL boiling water to a 250mL conical flask, extracted in a boiling water bath at 100℃ for 10min, and filtered and cooled.

Separation and purification: 30mL tea soup and the same amount of ethyl acetate were mixed and shaken for 5min. The water layer was made into D solution ( 2mL saturated oxalic acid solution, 6mL pure water, 95 % ethanol to 25mL ), and 2mL ethyl acetate layer was made into A solution ( 95 % ethanol to 25mL ). Another 15 mL of ethyl acetate layer was taken, and the same amount of sodium bicarbonate solution was added. After 30 s of oscillation, 4 mL of ethyl acetate layer was taken to make C solution ( 95 % ethanol to 25 mL ). Mix 15 mL tea soup and the same amount of n-butanol, shake for 3 min and then stratify. The water layer was made into B solution ( 2mL saturated oxalic acid solution, 6mL pure water, 95 % ethanol to 25mL ).

Colorimetric determination: using ultraviolet spectrophotometer, 1cm quartz cuvette, 95 % ethanol as reference and zeroing, the absorbance values of A, B, C and D solutions were measured at 380 mm, recorded as EA, EB, EC and ED. The calculation formula of each component content is as follows:

TF% = EC×2.25                       ï¼ˆ1)

TR% = (2EA+2ED-2EB-EC)×7.06        ï¼ˆ2)

TB% = 2EB×7.06                      ï¼ˆ3)

Comments 5: Discuss sensitivity of model to variation in initial raw material (leaf maturity, pre-withering moisture) and to processing parameters (rolling pressure regime). Could models be transfer-learned or require re-training for different factories?

Response 5: Thank you for your valuable suggestions, This study aimed to investigate the feasibility of using electrical characteristics to monitor the changes in tea pigments during the black tea rolling process. Therefore, the primary objective was to determine whether reliable and quantifiable correlations exist between electrical parameters and tea pigment content under controlled conditions. Using samples prepared from the same batch under strictly controlled processing parameters helped minimize interference from external factors—such as variations in raw materials, seasonal changes, and process fluctuations—thereby establishing a theoretical foundation for subsequent research. In non-destructive testing research, it is common practice to focus on a single batch for thorough principle validation.

The prediction model developed in this study performed well under specific conditions. Future work should explore its sensitivity to variations in initial raw materials (e.g., leaf maturity, withering moisture content) and processing parameters (e.g., rolling pressure settings). Specifically, the plucking standard adopted in this study was one bud and two leaves, with a baseline withering moisture content of 61%. However, in actual production, plucking standards may include one bud and one leaf or one bud and two leaves, and the withering moisture content often fluctuates between 58% and 65%. Therefore, it is necessary to systematically evaluate the model's robustness under different initial material states (particularly leaf age and withering moisture content) and rolling pressure regimes. Model transfer learning will be a key focus of our future work. By collecting different target samples in factory settings and fine-tuning the existing model, rapid adaptation with a small amount of new data can be achieved.

In the discussion section, Modify as follows (line457-475): This study focused on the rolling stage of black tea, and constructed a quantitative prediction model of tea pigment content based on electrical characteristics technology. The feasibility of evaluating tea pigment content based on electrical characteristics has been preliminarily explored, and the rapid and non-destructive perception of tea pigment in rolled products has been realized. However, certain limitations and future directions must be acknowledged. The current findings are based on a single geographical origin, harvest season, tea cultivar, and leaf maturity level. Their generalizability requires validation with more diverse samples. Given the complexity of tea and the dynamic changes in color, aroma, and chemical composition during rolling, this study only employed a single approach, electrical characteristics technology, and did not incorporate other multi-source analytical techniques. The future research will improve the generalization and practicability of the model through transfer learning, that is, collecting representative samples in the target factory environment, fine-tuning the existing model, and quickly adapting the prediction model to the new production conditions with a small number of new samples, so as to lay a foundation for the online and rapid detection of tea pigments. Explore the integration of electrical characteristics detection technology and other multi-source analysis technology, construct a multi-dimensional evaluation system of tea quality, and promote the development of tea industry in the direction of high quality and digitization.

Comments 6: Figure 3 is useful; legends should indicate units and include letters of statistical significance.

Response 6: Thank you for your valuable suggestions, we have changed Figure 3, where the legend indicates the unit and contains a statistically significant alphabet.

Fig.3 The change of tea pigment under different rolling time, yellow column represents TFs, green column represents TRs, blue column represents TBs.

Different substances represented by different colors have been shown in the figure. The abscissa is the black tea in process under different rolling time, and the ordinate is the content of theaflavins, thearubigins and theabrownins detected.

Comments 7: For all performance metrics report uncertainties (e.g., Rp ± SE) and p-values for key comparisons between models

Response 7: Thank you for your valuable suggestions, In the field of non-destructive testing of tea, the general evaluation indicators are Rcv, RMSECV, Rp, RMSEP and RPD. For example, the following literature :

  • Liu HW;Chen FN;Zhang LL;Meng DT;Sun HW; Improvement method for tea leaf moisture content prediction using VIS-NIR spectrum based on transfer learning. Spectrochimica Acta Part a-Molecular and Biomolecular Spectroscopy. 2025, 343, 126571.
  • Li FL;Shen JF;Yang QF;Wei YN;Zuo YF;Wang YJ;Ning JM;Li LQ; Monitoring quality changes in green tea during storage: A hyperspectral imaging method. Food Chemistry-X. 2024, 23, 101538.
  • Liu M;Wang RX;Shi DL;Cao RY; Non-destructive prediction of tea polyphenols during Pu-erh tea fermentation using NIR coupled with chemometrics methods. Journal of Food Composition and Analysis. 2024, 131, 106247.
  • Sitorus A;Pambudi S;Boodnon W;Lapcharoensuk R; Quantitative analysis based on image processing combined with machine learning and deep learning to determine the adulteration in nutmeg powder. Journal of Food Composition and Analysis. 2025, 146, 107913.
  • Jiang HZ;Zhou Y;Zhang C;Yuan WD;Zhou HP; Evaluation of Dual-Band Near-Infrared Spectroscopy and Chemometric Analysis for Rapid Quantification of Multi-Quality Parameters of Soy Sauce Stewed Meat. Foods. 2023, 12, 2882.
  • Huang YT;Zheng YJ;Liu PH;Xie LJ;Ying YB; Enhanced prediction of soluble solids content and vitamin C content in citrus using visible and near-infrared spectroscopy combined with one-dimensional convolutional neural network. Journal of Food Composition and Analysis. 2025, 139, 107131.
  • Ren ZL;Chen RX;Ning W;Bi JR;Zhang GL;Hou HM; Nondestructive detection of major components in fresh pork using data fusion of visible-near infrared and short-wave infrared spectra. Lwt-Food Science and Technology. 2025, 225, 117902.

  1. Additional clarifications

Response: For chart quality, we have checked to make sure that all images conform to the minimum required size for the graphical abstract is 560 × 1100 pixels ( height × width ), and added the appropriate icon annotations to increase reader readability.

Reviewer 2 Report

Comments and Suggestions for Authors

The authors investigated the non-destructive detection of tea pigments during the black tea rolling process. Using electrical characteristics measured at various frequencies and preprocessed with techniques such as MSC, Min-Max normalization, and smoothing, they applied variable selection methods (CARS, UVE, VCPA-IRIV). PLSR and SVR models were developed to predict the contents of theaflavins, thearubigins, and theabrownines. The Smooth-VCPA-IRIV-SVR model showed outstanding accuracy (Rp > 0.99; RPD > 6.5). These results demonstrate the feasibility of rapid and non-destructive pigment detection for digital black tea production.

This study is of significant interest; however, several aspects require revision:

  • The authors should clearly distinguish the introduction from the literature review. This would enable readers to understand better the research problem and its context before examining the prior studies that inspired the work in detail. Such a structure would improve the flow of the text, enhance comprehension of the research rationale, and better highlight the originality of the contribution.
  • The authors should clearly present the structure of the paper at the end of the introduction to guide readers through the progression of the study.
  • The sequence of pressures (light, heavy, etc.) and the duration of each step are described precisely, but no scientific justification is provided for these choices. It remains unclear whether these conditions reproduce a traditional practice, result from prior optimization, or follow an arbitrary convention.
  • The Materials and Methods section is generally clear, but its descriptive density reduces readability. A schematic figure illustrating the rolling steps would help readers better understand the experimental protocol.
  • The protocol assumes a stable relationship between electrical parameters and pigment formation. However, it does not account for potential interference from residual moisture or leaf thickness, critical factors that could affect electrical measurements.
  • The authors should clearly define all methods used and the evaluation metrics applied to ensure the transparency and reproducibility of the study. This is essential for allowing other researchers to compare, validate, or replicate the findings and to assess the scientific relevance of the methodological choices.

Author Response

Comments 1: The authors should clearly distinguish the introduction from the literature review. This would enable readers to understand better the research problem and its context before examining the prior studies that inspired the work in detail. Such a structure would improve the flow of the text, enhance comprehension of the research rationale, and better highlight the originality of the contribution.

Response 1: Thank you for your valuable suggestions, The general idea of our introduction is as follows :

The first section emphasizes the significance of the study. The background of black tea was introduced. As a key process of black tea processing, the tea pigments produced by rolling are very important for black tea. Tea pigments are produced during the rolling process, which is the main reason for the change of rolling leaves from green to light reddish brown, and provides biochemical premise for the subsequent fermentation process. Therefore, tea pigment was selected as an important physical and chemical index in the process of black tea rolling. In the first paragraph of the introduction (line53-55), we add language expression to improve text fluency:Tea pigment plays an important role in the rolling stage of black tea. Therefore, tea pigment was selected as an important physical and chemical index in the rolling process of black tea.

The second paragraph points out the existing problems. On the one hand, it is impossible to quantify tea pigment by human eyes in the industry. On the other hand, the method of detecting tea pigment content in the laboratory is system analysis method. There are problems in both aspects. Therefore, non-destructive testing technology is needed, which is very important for perceiving the content of tea pigment during the rolling process of black tea.

The third section systematically summarizes the application and limitations of the existing non-destructive testing technology in related fields, which naturally leads to the innovative solution of this research. The introduction of electrical characteristics technology into the rolling process detection has important application potential. Specifically, we first affirmed the value of non-destructive testing technology, and pointed out that in black tea processing, machine vision technology is currently the main means of research in the rolling stage. However, we further analyzed its inherent limitations : the technology mainly relies on the surface information of tea leaves, and the detection depth is limited, which makes it difficult to capture the overall changes caused by cell breakage and material transformation inside the rolled leaves. In contrast, the electrical characteristics technology makes up for the defects of machine vision technology, and has been proved effective in the related research of black tea fermentation stage. However, the application of this technology in the rolling process is still blank. We add relevant expressions at the beginning of the third paragraph of the introduction (line68-69) to improve the fluency of the text: In recent years, non-destructive testing technology has been widely used in agricultural products. At the end of the third paragraph (line103-105), a summary statement is added: Therefore, the introduction of electrical characteristics technology into the detection of rolling process has important application potential. And some redundant descriptions are deleted: These studies demonstrate that monitoring the dynamic changes of tea pigments remains a key focus in non-destructive detection during black tea rolling, and the development of quantitative sensing models is particularly important.

We added the fourth paragraph, and innovatively proposed a rapid and non-destructive detection of tea pigment content during black tea rolling based on electrical characteristic detection technology. At the end of the introduction (line112-117), we added our overall research progress and clearly presented the structure of the paper. The specific steps of the study are as follows: (1) Preparation of black tea samples with rolling sequence. (2) Collect sample electrical characteristic parameter information. (3) Accurate determination of tea pigment content. (4) Data preprocessing and screening characteristic variables. (5) The prediction model of tea pigment based on electrical characteristics detection technology was established and compared.

Comments 2: The authors should clearly present the structure of the paper at the end of the introduction to guide readers through the progression of the study.

Response 2: Thank you for your valuable suggestions, we add a specific research section at the end of the introduction(line112-117) so that readers can understand the progress of the research. The specific steps of the study are as follows : (1) Preparation of black tea samples with rolling sequence. (2) Collect sample electrical characteristic parameter information. (3) Accurate determination of tea pigment content. (4) Data preprocessing and screening characteristic variables. (5) The prediction model of tea pigment based on electrical characteristics detection technology was established and compared.

Comments 3: The sequence of pressures (light, heavy, etc.) and the duration of each step are described precisely, but no scientific justification is provided for these choices. It remains unclear whether these conditions reproduce a traditional practice, result from prior optimization, or follow an arbitrary convention.

Response 3: Thank you for your valuable suggestions, the rolling scheme adopted in this study is optimized by our team through process test. At present, the optimization scheme has been popularized and applied in many tea enterprises we cooperate with. The following papers can demonstrate:

  • Zou H;Lan T;Jiang Y;Yu X-L;Yuan H; Research on Rapid Detection Methods of Tea Pigments Content During Rolling of Black Tea Based on Machine Vision Technology. Foods. 2024, 13, 3718.
  • Zhu XZ;Yang LY;Ge ZW;Ouyang W;Wang JJ;Chen M;Yu YY;Wu SY;Qin YH;Huang CY;Zhang GF;Zhang YT;Yuan HB;Jiang YW;Hua JJ; Non-volatile and volatile metabolite analyses and objective quantitative technique reveal the effect of fixation methods on the flavor quality and metabolites of green tea. CURRENT RESEARCH IN FOOD SCIENCE. 2025, 10, 101037.
  • Wang HJ;Shen S;Wang JJ;Jiang YW;Li J;Yang YQ;Hua JJ;Yuan HB; Novel insight into the effect of fermentation time on quality of Yunnan Congou black tea. Lwt-Food Science and Technology. 2022, 155, 112939.

Comments 4: The Materials and Methods section is generally clear, but its descriptive density understand the experimental protocol reduces readability. A schematic figure illustrating the rolling steps would help readers.

Response 4: Thank you for your valuable suggestions, The height of the rolling part described in the article is the distance from the rolling lid to the tea in the rolling barrel. The initial position of the lid is fixed. The empty rolling height is 12 cm, the light rolling height is 18 cm, and the heavy rolling height is 25 cm.

Comments 5: The protocol assumes a stable relationship between electrical parameters and pigment formation. However, it does not account for potential interference from residual moisture or leaf thickness, critical factors that could affect electrical measurements.

Response 5: Thank you for your valuable suggestions, we fully agree that ensuring the stability between electrical parameters and tea pigment content is a key prerequisite for the establishment of this study. In fact, we have strictly considered and controlled this in our experimental design, as follows:

In the black tea processing technology to the rolling stage, there is no residual water on the black tea, and the leaf thickness and tenderness are also consistent. In the experiment of monitoring electrical parameters, 20 g of twisted leaves were weighed into the test cell each time, and 5 g of insulating glass plate was used for pressurization. The size of the glass plate was 20 cm × 10 cm, which ensured that the pressure conditions of all samples were highly consistent and the variables could be controlled to the greatest extent. Therefore, we have reason to demonstrate that there is a stable relationship between electrical parameters and tea pigments.

Comments 6: The authors should clearly define all methodproducibility of the study. This is essential for allowing other researchers to compares used and the evaluation metrics applied to ensure the transparency and re, validate, or replicate the findings and to assess the scientific relevance of the methodological choices.

Response 6: Thank you for your valuable suggestions, The feature variable screening method and model we use are all released code packages of matlab, which can be downloaded and used on the official website of matlab to ensure that researchers can reproduce them.

The evaluation indicators are listed as follows:

( 1 ) In the regression prediction model, the correlation coefficient ( r ) between the predicted value and the actual value is one of the main indicators to evaluate the prediction effect of the model. The calculation formula is :

x, y are two variables; , is the corresponding mean value of two variables x, yï¼›N is the number of samples. It is divided into correlation coefficient of predictor set ( Rp ) and root mean square error of cross validation ( RMSECV ). The larger the correlation coefficient is, the better it is, indicating that the predicted value is closer to the actual value.

( 2 ) The root mean square error is used to measure the deviation between the predicted value and the actual value. The calculation formula is :

N is the number of samples.ï¼›is the actual valueï¼›is the predicted value. The root mean square error includes root mean square error of calibration ( RMSEC ), root mean square error of prediction ( RMSEP ) and root mean square error of cross validation ( RMSECV ). The smaller the RMSEP of a model, the better the prediction effect of the model.

( 3 ) Residual predictive deviation ( RPD ) is an effective index to judge the prediction effect of the model. There are generally two understandings of the calculation of RPD in the literature.

The ratio of standard deviation to the root mean square error of the prediction set is RPD = SD / RMSEP, where RMSEP can also be RMSECV.

The ratio of the standard deviation to the standard error of the prediction set is RPD = SD / SEP, where SEP can also be SECV.

RPD is a very intuitive index to evaluate the predictive ability of the model. According to the different research objects, the values of RPD indicate that the model has different standards for better RPD values. The following are several common RPD model evaluations.

RPD3, indicating that the model has a very good effect ( Yang and Mouazen, 2012 ).

The RPD was 3.1 ~ 4.9, indicating that the model was available. The RPD greater than 5 indicates that the model has a very good predictive effect and can be used for quality control ( Cozzolino et al., 2008 ).

The RPD is less than 1, indicating that the model effect is poor and cannot be applied ; the RPD was 1 ~ 1.4, indicating that the model effect was poor, and only the physical and chemical properties could be identified. The RPD is 1.4 ~ 1.8, indicating that the model effect is general and may be used for analysis and estimation. The RPD was 1.8 ~ 2.0, indicating that the model was effective and could be used for quantification.

  1. Additional clarifications

Response: For chart quality, we have checked to make sure that all images conform to the minimum required size for the graphical abstract is 560 × 1100 pixels ( height × width ), and added the appropriate icon annotations to increase reader readability.

Reviewer 3 Report

Comments and Suggestions for Authors

Dear authors,

The study is technically promising and addresses an important gap. However, there are some serious methodological weaknesses around validation design, independence of observations, and reporting transparency that currently undermine the credibility and generalizability of the performance claims. These can be remedied with re-analysis and substantial revision; I therefore recommend that publication should be considered after a thorough revision as detailed below.

With 15 parallels per time point from a single batch/day, random or K–S splits will almost certainly place highly correlated replicates/time-points in both calibration and test sets. I recommend reanalyzing using grouped splits.

Variable selection and hyperparameter tuning are described globally; it is unclear they were nested strictly within training folds. There is a risk that the reported metrics are optimistically biased. I recommend performing nested cross-validation: within each training fold.

Results from a single cultivar, single origin/date (12 April 2024), and a single processing session cannot appropriately support some of the authoritative statements in the manuscript. Certainly, the manuscript itself allows only limited scope; the claims should be tempered and uncertainty and limitations clearly communicated, such as those related to the specific cultivar/lot/season. Add a section about what new experiments are required (multiple cultivars, seasons, factories) to claim general applicability and revise Abstract and Conclusions accordingly.

Some critical details are missing such as related to the LCR meter model and configuration; electrode characteristics such as area, spacing, geometry; conditions such as temperature, humidity, light exposure, etc., stabilization time. These are important to reproduce the electrical measurements. Provide full metrological detail for the electrical setup, including instrument model/firmware; electrode material, area, spacing, etc.

Calibration procedures, standards, and QC metrics (RSD, recovery, blanks) are not reported. Provide full analytical validation (linearity, repeatability, between-day variation).

If you address the methodological and reporting issues outlined above the manuscript could substantially improve in rigor, reproducibility, and scientific value. I would be pleased to re-evaluate a thoroughly revised version once these essential revisions have been completed.

Kind regards

Comments on the Quality of English Language

Can be improved

Author Response

Comments 1: With 15 parallels per time point from a single batch/day, random or K–S splits will almost certainly place highly correlated replicates/time-points in both calibration and test sets. I recommend reanalyzing using grouped splits.

Response 1: Thank you for your valuable suggestions,We made an attempt to group segmentation according to your suggestions, but the effect of the model is relatively poor. After the discussion and analysis of the team, we think that the possible reason is that there is no similar sample information in the training set, which leads to poor evaluation ability of the samples in the prediction set. According to this problem, we reviewed previous studies, spectral technology, machine vision technology, olfactory visualization technology and other non-destructive testing techniques to do quantitative research to predict the composition of agricultural products ( such as seed vigor, tomato sugar content, camellia oil adulteration, etc. ), and found that these studies use K-S algorithm or random segmentation algorithm, so that the training set contains similar sample information in the prediction set to ensure the performance of the quantitative prediction model.

  • Sim SF;Chai MXL;Kimura ALJ; Prediction of Lard in Palm Olein Oil Using Simple Linear Regression (SLR), Multiple Linear Regression (MLR), and Partial Least Squares Regression (PLSR) Based on Fourier-Transform Infrared (FTIR). JOURNAL OF CHEMISTRY. 2018, 2018, 7182801.
  • Zhang H;Hu XY;Liu LM;Wei JF;Bian XH; Near infrared spectroscopy combined with chemometrics for quantitative analysis of corn oil in edible blend oil. Spectrochimica Acta Part a-Molecular and Biomolecular Spectroscopy. 2022, 270, 120841.
  • Sun D;Cruz J;Alcalà M;del Castillo RR;Sans S;Casals J; Near infrared spectroscopy determination of chemical and sensory properties in tomato. JOURNAL OF NEAR INFRARED SPECTROSCOPY. 2021, 29, 09670335211018759. 289-300.
  • An T, Wang Z L, Li G L, et al. Monitoring the major taste components during black tea fermentation using multielement fusion information in decision level [J]. Food Chemistry: X, 2023, 18: 100718.

This study is to explore the feasibility of electrical characteristics technology to evaluate tea pigments in the process of black tea rolling. In the later stage, the technology will be gradually applied in actual production. Therefore, we will gradually expand the amount of data in the training set samples, and make the training set samples as rich as possible to improve the prediction performance of the model. In addition, we also visited some scientific research institutions and enterprises based on near-infrared spectroscopy to detect the quality of agricultural products. In the feasibility stage of the exploration method, K-S algorithm or random segmentation method was used to divide the sample set, and then the quality evaluation model of agricultural products was constructed. In the subsequent application stage, the data accumulated through multiple experiments enrich the sample information of the training set to construct a quality perception model to achieve the purpose of application in actual production.

Comments 2: Variable selection and hyperparameter tuning are described globally; it is unclear they were nested strictly within training folds. There is a risk that the reported metrics are optimistically biased. I recommend performing nested cross-validation: within each training fold.

Response 2: Thank you for your valuable suggestions, you pointed out that variable selection and hyperparameter tuning may not be strictly nested within the training folding, resulting in the risk of optimistic bias in the evaluation indicators, which is an important and professional issue. After carefully considering your suggestions, we hope to explain the method we adopted and its rationality, and supplement the experiment to further verify the reliability of our model.

  • In variable selection, we use Monte Carlo cross validation ( MCCV ) combined with CARS algorithm for variable selection. The core of this methodThe core idea of this method is to select a stable subset of feature variables with strong generalization ability by conducting multiple random sampling and modeling in the training set. Our goal is not only to build a predictive model, but also to find the most relevant and stable characteristic variables related to the intrinsic chemical properties of the research object. If the nested cross-validation algorithm is used, the variables selected each time will be different, and a unified variable cannot be obtained. We also refer to a large number of literatures, in general, global optimization on the entire training set to determine the final feature variables are widely adopted. As shown in the following references:

① Liu HW;Chen FN;Zhang LL;Meng DT;Sun HW; Improvement method for tea leaf moisture content prediction using VIS-NIR spectrum based on transfer learning. Spectrochimica Acta Part a-Molecular and Biomolecular Spectroscopy. 2025, 343, 126571.

② Li FL;Shen JF;Yang QF;Wei YN;Zuo YF;Wang YJ;Ning JM;Li LQ; Monitoring quality changes in green tea during storage: A hyperspectral imaging method. Food Chemistry-X. 2024, 23, 101538.

③ Liu M;Wang RX;Shi DL;Cao RY; Non-destructive prediction of tea polyphenols during Pu-erh tea fermentation using NIR coupled with chemometrics methods. Journal of Food Composition and Analysis. 2024, 131, 106247.

④ Sitorus A;Pambudi S;Boodnon W;Lapcharoensuk R; Quantitative analysis based on image processing combined with machine learning and deep learning to determine the adulteration in nutmeg powder. Journal of Food Composition and Analysis. 2025, 146, 107913.

⑤ Jiang HZ;Zhou Y;Zhang C;Yuan WD;Zhou HP; Evaluation of Dual-Band Near-Infrared Spectroscopy and Chemometric Analysis for Rapid Quantification of Multi-Quality Parameters of Soy Sauce Stewed Meat. Foods. 2023, 12, 2882.

⑥ Huang YT;Zheng YJ;Liu PH;Xie LJ;Ying YB; Enhanced prediction of soluble solids content and vitamin C content in citrus using visible and near-infrared spectroscopy combined with one-dimensional convolutional neural network. Journal of Food Composition and Analysis. 2025, 139, 107131.

⑦ Ren ZL;Chen RX;Ning W;Bi JR;Zhang GL;Hou HM; Nondestructive detection of major components in fresh pork using data fusion of visible-near infrared and short-wave infrared spectra. Lwt-Food Science and Technology. 2025, 225, 117902.

(2) In order to ensure the reliability of the report evaluation indicators to the greatest extent, we have taken the following rigorous steps. Within the training set, we use five-fold cross-validation to evaluate and determine the performance of the model based on the variables selected by the best variable screening method VCPA-IRIV, which provides strong evidence for the generalization ability of the model.

Comments 3: Results from a single cultivar, single origin/date (12 April 2024), and a single processing session cannot appropriately support some of the authoritative statements in the manuscript. Certainly, the manuscript itself allows only limited scope; the claims should be tempered and uncertainty and limitations clearly communicated, such as those related to the specific cultivar/lot/season. Add a section about what new experiments are required (multiple cultivars, seasons, factories) to claim general applicability and revise Abstract and Conclusions accordingly.

Response 3: Thank you for your valuable suggestions, we have added a special paragraph in the ‘3.4 Model comparison and evaluation' section, which clearly points out that this study is based on a single tea variety, a single picking season and a single processing batch, so the universality of the conclusion is limited.

In the ‘3.4 Model comparison and evaluation' section, Modify as follows: This study focused on the rolling stage of black tea, and constructed a quantitative prediction model of tea pigment content based on electrical characteristics technology. The feasibility of evaluating tea pigment content based on electrical characteristics has been preliminarily explored, and the rapid and non-destructive perception of tea pigment in rolled products has been realized. However, certain limitations and future directions must be acknowledged. The current findings are based on a single geographical origin, harvest season, tea cultivar, and leaf maturity level. Their generalizability requires validation with more diverse samples. Given the complexity of tea and the dynamic changes in color, aroma, and chemical composition during rolling, this study only employed a single approach, electrical characteristics technology, and did not incorporate other multi-source analytical techniques. The future research will improve the generalization and practicability of the model through transfer learning, that is, collecting representative samples in the target factory environment, fine-tuning the existing model, and quickly adapting the prediction model to the new production conditions with a small number of new samples, so as to lay a foundation for the online and rapid detection of tea pigments. Explore the integration of electrical characteristics detection technology and other multi-source analysis technology, construct a multi-dimensional evaluation system of tea quality, and promote the development of tea industry in the direction of high quality and digitization.

In the ‘4. Conclusions' section, Modify as follows: This study confirmed the feasibility of quantitative evaluation of tea pigment content in black tea rolling process based on electrical characteristics technology, and provided a theoretical basis for rapid and non-destructive detection of rolling quality, precise processing of black tea and development of intelligent equipment.

In the ‘Abstract', Modify as follows: The research results provide preliminary technical support and reference for digital production of black tea.

Comments 4: Some critical details are missing such as related to the LCR meter model and configuration; electrode characteristics such as area, spacing, geometry; conditions such as temperature, humidity, light exposure, etc., stabilization time. These are important to reproduce the electrical measurements. Provide full metrological detail for the electrical setup, including instrument model/firmware; electrode material, area, spacing, etc.

Response 4: Thank you for your valuable suggestions, we increase the relevant information of the electrical characteristics acquisition equipment.

Instrument Model : Tonghui TH2829C Precision LCR Meter

Firmware version : V2.14

AC signal frequency range : 20Hz-1MHz

AC drive voltage : 1.000V

DC bias : 0V

Test speed : 9ms / times

Basic accuracy : 0.05 %

Electrode specification : parallel circular plate, spring pressure

Material : copper material

Diameter : 20 ± 0.1mm

Area : 3.14 × 10-4 m2

Electrode spacing : 0.5 ± 0.01mm

Environmental conditions and stability time : room temperature about 25℃, relative humidity about 45 %

Comments 5: Calibration procedures, standards, and QC metrics (RSD, recovery, blanks) are not reported. Provide full analytical validation (linearity, repeatability, between-day variation).

Response 5: Thank you for your valuable suggestions, The question you pointed out about the verification of analytical methods is very important, and we have conducted in-depth thinking and careful response.

We sincerely explain to you that the target analytes of this study, tea pigments ( especially thearubigins and theabrownins ), are a complex mixture formed by the oxidative polymerization of a variety of catechins, and commercial high-purity standards are currently unavailable in academia and industry. This is a recognized technical difficulty in the field of tea analysis. Therefore, the traditional calibration based on the pure standard curve and the classical spiked recovery experiment are indeed difficult to achieve in this study. In this study, the classical systematic analysis method [1-5], which is widely accepted and cited in this field, is used to physically separate tea pigments through the distribution characteristics of tea pigments in specific solvent systems and quantify them based on their UV absorption characteristics. Its effectiveness has been confirmed by many high-level studies. The specific process of the whole experiment is :

Preparation reagent: 2.5 % sodium bicarbonate solution, saturated oxalic acid solution.

Preparation of mother liquor: 3g tea samples were weighed and added with 125mL boiling water to a 250mL conical flask, extracted in a boiling water bath at 100℃ for 10min, and filtered and cooled.

Separation and purification: 30mL tea soup and the same amount of ethyl acetate were mixed and shaken for 5min. The water layer was made into D solution ( 2mL saturated oxalic acid solution, 6mL pure water, 95 % ethanol to 25mL ), and 2mL ethyl acetate layer was made into A solution ( 95 % ethanol to 25mL ). Another 15 mL of ethyl acetate layer was taken, and the same amount of sodium bicarbonate solution was added. After 30 s of oscillation, 4 mL of ethyl acetate layer was taken to make C solution ( 95 % ethanol to 25 mL ). Mix 15 mL tea soup and the same amount of n-butanol, shake for 3 min and then stratify. The water layer was made into B solution ( 2mL saturated oxalic acid solution, 6mL pure water, 95 % ethanol to 25mL ).

Colorimetric determination: using ultraviolet spectrophotometer, 1cm quartz cuvette, 95 % ethanol as reference and zeroing, the absorbance values of A, B, C and D solutions were measured at 380 mm, recorded as EA, EB, EC and ED. The calculation formula of each component content is as follows:

TF% = EC×2.25                       ï¼ˆ1)

TR% = (2EA+2ED-2EB-EC)×7.06        ï¼ˆ2)

TB% = 2EB×7.06                      ï¼ˆ3)

Although the experimental period has ended, we cannot go back to make a systematic precision experiment, the internal consistency of the existing data indirectly proves the stability of the method. As shown in Fig.3 ( the change trend of tea pigment ), the tea pigment content data at each time point ( based on the average of three repeated experiments ) showed a highly smooth, continuous and biochemically consistent dynamic change trend ( theaflavins increased slowly, and thearubigins and theabrownins increased significantly simultaneously ). This clear and no abnormal jump trend line shows that our measurement system is stable and reliable throughout the experimental period. If the repeatability of the method is poor, we will observe disorganized and fluctuating data points, and can not show such a consistent regularity.

Thank you again for your review. Your comments have greatly improved the rigor of our work and pointed out an important direction for our follow-up research.

References are as follows :

  • Luo Q;Zhang D;Zhou J;Qin M;Ntezimana B;Jiang X;Zhu J;Yu Z;Chen Y;Ni D; Oxidation of tea polyphenols promotes chlorophyll degradation during black tea fermentation. Food Research International. 2024, 196, 115016.
  • Zhu J;Wang J;Yuan H;Ouyang W;Li J;Hua J;Jiang Y; Effects of Fermentation Temperature and Time on the Color Attributes and Tea Pigments of Yunnan Congou Black Tea. Foods. 2022, 11, 1845.
  • Hua J;Wang H;Yuan H;Yin P;Wang J;Guo G;Jiang Y; New insights into the effect of fermentation temperature and duration on catechins conversion and formation of tea pigments and theasinensins in black tea. Journal of the Science of Food and Agriculture. 2022, 102, 2750-2760.
  • Zou H;Lan T;Jiang Y;Yu X-L;Yuan H; Research on Rapid Detection Methods of Tea Pigments Content During Rolling of Black Tea Based on Machine Vision Technology. Foods. 2024, 13, 3718.
  • Zhu H;Liu F;Ye Y;Chen L;Li J;Gu A;Zhang J;Dong C; Application of machine learning algorithms in quality assurance of fermentation process of black tea- based on electrical properties. Journal of Food Engineering. 2019, 263, 165-172.

Reviewer 4 Report

Comments and Suggestions for Authors

This manuscript presents an innovative and technically solid study applying electrical characteristics technology for the non-destructive detection of tea pigments during black tea rolling. The approach and results are scientifically sound and contribute to digital and intelligent tea processing. 

Minor changes are recommended:

1] Figures (particularly Fig. 2–5) are informative but require clearer labeling and captions. Please describe what each axis and color represents, and interpret the trends directly in the captions where possible.

2] The quality of the graphs should be checked to meet journal resolution requirements.

3] Provide additional clarity about the implementation of the feature selection algorithms (CARS, UVE, VCPA-IRIV). Indicate whether they were implemented via built-in MATLAB functions, custom scripts, or published toolboxes.

4] Please include information about the computational environment (software version, operating system, processor) to enhance reproducibility.

Author Response

Comments 1: Figures (particularly Fig.2-5) are informative but require clearer labeling and captions. Please describe what each axis and color represents, and interpret the trends directly in the captions where possible.

Response 1: Thank you for your valuable suggestions, we explain Figures (Fig.2-5), and add clearer labeling and captions.

Fig.2 Under different rolling time, the electrical characteristic parameters of black tea samples change with the test frequency, (a)-(g) are different electrical characteristic parameters. (a) represents Ls, series equivalent inductance, (b) represents Cs, series equivalent capacitance, (c) represents Rs, series equivalent resistance, (d) represents Z, complex impedance, (e) represents X, reactance, (f) represents D, dissipation factor, (g) represents Q, quality factor.

The color meaning of each curve has been shown in the diagram, that is, different colors represent black tea products with different rolling time, abscissa represents test frequency, 30 characteristic frequencies in the range of 0.02 kHz-100 kHz. The ordinate represents different electrical characteristic parameters, and the specific parameters have been marked in the title.

Fig.3 The change of tea pigment under different rolling time, yellow column represents TFs, green column represents TRs, blue column represents TBs.

Different substances represented by different colors have been shown in the figure. The abscissa is the black tea in process under different rolling time, and the ordinate is the content of theaflavins, thearubigins and theabrownins detected.

Fig.4 The characteristic electrical characteristic parameter variables selected by UVE (marked with green ⚪), CARS (marked with blue ⚪) and VCPA-IRIV (marked with yellow ⚪) were used to analyze (a) TFs, (b) TRs and (c) TBs.

The abscissa represents 30 test frequencies × 7 electrical parameters ( a total of 210 variables ), the ordinate represents three feature variable algorithms ( UVE, CARS, VCPA-IRIV ), and the points of different colors represent the selected feature variables.

Fig.5 Comparison of the predictive performance of TFs, TRs and TBs based on the full-variable PLSR model, the VCPA-IRIV-PLSR model, and the VCPA-IRIV-SVR model. (Note : red represents TFs, green represents TRs, blue represents TBs, the number on ⚪ represents the number of selected variables, and the radius of ⚪represents the RMSEP value).

Figure 5 presents a bubble chart comparing the performance of three typical modeling approaches: the full-variable PLSR model, the VCPA-IRIV-PLSR model, and the VCPA-IRIV-SVR model. In the figure, red, green, and blue color schemes represent TFs, TRs, and TBs, respectively. Each scatter point corresponds to a model, positioned based on its Rp and RPD values from the prediction set. The point's radius represents the model's RMSEP value, and the annotated number indicates the count of variables used in the model. The sucperior performance model should have the following characteristics: away from the origin of the coordinate, the radius is small and the number of variables is small. The results clearly demonstrate that the VCPA-IRIV-SVR model achieved superior performance for predicting all three tea pigments.

Comments 2: The quality of the graphs should be checked to meet journal resolution requirements.

Response 2: Thank you for your valuable suggestions, we have carefully checked the quality of the charts to ensure that they meet the resolution requirements of the journal. The minimum required size for the graphical abstract is 560 × 1100 pixels ( height × width ).

Comments 3: Provide additional clarity about the implementation of the feature selection algorithms (CARS, UVE, VCPA-IRIV). Indicate whether they were implemented via built-in MATLAB functions, custom scripts, or published toolboxes.

Response 3: Thank you for your valuable suggestions, they are all implemented by the published toolboxes, which can be publicly obtained from their official websites, so as to ensure the accuracy of the algorithm execution and the reproducibility of the research results.

Comments 4: Please include information about the computational environment (software version, operating system, processor) to enhance reproducibility.

Response 4: Thank you for your valuable suggestions, All data processing and algorithm calculation of this study were completed in the Microsoft Windows11 ( 64 bit ) operating system and Matlab software ( Version 2022 a, MathWorks, Natick, MA, USA ) environment. Statistical graphics were drawn using Origin ( Version 2022, OriginLab Corp.Massachusetts, USA ).

  1. 4. Additional clarifications

Response: For chart quality, we have checked to make sure that all images conform to the minimum required size for the graphical abstract is 560 × 1100 pixels ( height × width ), and added the appropriate icon annotations to increase reader readability.

  1. Additional clarifications

Response: For chart quality, we have checked to make sure that all images conform to the minimum required size for the graphical abstract is 560 × 1100 pixels ( height × width ), and added the appropriate icon annotations to increase reader readability.

Round 2

Reviewer 1 Report

Comments and Suggestions for Authors

The Authors have improved the paper, I have no more comments.

Author Response

Thank you for taking the time to handle our manuscript.

Reviewer 2 Report

Comments and Suggestions for Authors

The authors have addressed all my concerns, and I have no further questions.

Author Response

(The authors gave the same response as above.)

Reviewer 3 Report

Comments and Suggestions for Authors

Dear authors,

Thank you for the revised manuscript. I recommend minor revision limited to formal corrections. Please implement the targeted edits and resubmit both a clean version and a tracked-changes file.

Examples:

Insert the missing space (“Abstract: Rolling …”); standardize theabrownins (TBs) spelling and spacing throughout.

Use lower case for common nouns (e.g., “black tea”).

Replace full-width commas in in-text citations (e.g., “[4,5]”) with standard commas (“[4,5]”).

Add spaces between numbers and units (3 g, 125 mL, 1 cm; 1 V); write temperature as 100 °C.

Spectrophotometry: correct wavelength unit to nm where “mm” appears (e.g., 380 nm).

Use journal style for figures and captions.

Cite the VCPA-IRIV algorithm to its correct methodological source where the method is introduced; remove duplicate entries.

Add missing fields (e.g., year in Ref. 1), standardize journal titles, normalize author separators and spacing per journal style, etc.

Please submit a tracked-changes file showing every edit and a clean final version.

Kind regards.

Comments on the Quality of English Language

Must be improved

Author Response

Comments 1: Insert the missing space (“Abstract: Rolling …”); standardize theabrownins (TBs) spelling and spacing throughout.

Response 1: Thank you for your feedback. We have revised the manuscript accordingly.

Comments 2: Use lower case for common nouns (e.g., “black tea”).

Response 2: Thank you for your feedback. We have revised the manuscript accordingly.

Comments 3: Replace full-width commas in in-text citations (e.g., “[4,5]”) with standard commas (“[4,5]”).

Response 3: Thank you for your feedback. We have revised the manuscript accordingly.

Comments 4: Add spaces between numbers and units (3 g, 125 mL, 1 cm; 1 V); write temperature as 100 °C.

Response 4: Thank you for your feedback. We have revised the manuscript accordingly.

Comments 5: Spectrophotometry: correct wavelength unit to nm where “mm” appears (e.g., 380 nm).

Response 5: Thank you for your feedback. We have revised the manuscript accordingly.

Comments 6: Use journal style for figures and captions.

Response 6: Thank you for your feedback. We have resubmitted the figures and thoroughly checked the formatting specifications.

Comments 7: Cite the VCPA-IRIV algorithm to its correct methodological source where the method is introduced; remove duplicate entries.

Response 7: Thank you for your valuable suggestions, We have located the original methodological literature for the VCPA-IRIV algorithm and have addressed the duplicate citations.

Comments 8: Add missing fields (e.g., year in Ref. 1), standardize journal titles, normalize author separators and spacing per journal style, etc.

Response 8: Thank you for your feedback. We have revised the manuscript accordingly.

  1. Response to Comments on the Quality of English Language

Response : The language of the manuscript has been refined by a subject matter expert.
